# Pathway activation model for personalized prediction of drug synergy

Quang Thinh Trac[1†], Yue Huang[2†], Tom Erkers[3], Päivi Östling[3,4], Anna Bohlin[5], Albin Osterroos[6], Mattias Vesterlund[3], Rozbeh Jafari[3], Ioannis Siavelis[3], Helena Backvall[3], Santeri Kiviluoto[3], Lukas Orre[3], Mattias Rantalainen[1], Janne Lehtiö[3], Soren Lehmann[5,6], Olli Kallioniemi[3,4], Yudi Pawitan[1], Trung Nghia Vu[1]*

[1]Department of Medical Epidemiology and Biostatistics, Karolinska Institutet, Stockholm, Sweden; [2]Department of Health Statistics, School of Public Health, Weifang Medical University, Weifang, China; [3]Department of Oncology Pathology, Karolinska Institutet, Science for Life Laboratory, Stockholm, Sweden; [4]Institute for Molecular Medicine Finland, University of Helsinki, Helsinki, Finland; [5]Department of Medicine Huddinge, Karolinska Institutet, Unit for Hematology, Karolinska University Hospital Huddinge, Stockholm, Sweden; [6]Department of Medical Sciences, Hematology, Uppsala University Hospital, Uppsala, Sweden

*For correspondence: trungnghia.vu@ki.se

[†]These authors contributed equally to this work

Competing interest: The authors declare that no competing interests exist.

## eLife Assessment

This **valuable** study presents a deep learning framework for predicting synergistic drug combinations for cancer treatment in the AstraZeneca-Sanger (AZS) DREAM Challenge dataset. The level of evidence seems **solid**, although performance on some datasets seems unconvincing and further validation would be required to demonstrate the generalizability of the model and, in turn, its clinical relevance. The reported tool, DIPx, could be of use for personalized drug synergy prediction and exploring the activated pathways related to the effects of drug combinations.

**Abstract** Targeted monotherapies for cancer often fail due to inherent or acquired drug resistance. By aiming at multiple targets simultaneously, drug combinations can produce synergistic interactions that increase drug effectiveness and reduce resistance. Computational models based on the integration of omics data have been used to identify synergistic combinations, but predicting drug synergy remains a challenge. Here, we introduce Drug synergy Interaction Prediction (DIPx), an algorithm for personalized prediction of drug synergy based on biologically motivated tumor- and drug-specific pathway activation scores (PASs). We trained and validated DIPx in the AstraZeneca-Sanger (AZS) DREAM Challenge human cell-line dataset using two separate test sets: Test Set 1 comprised the combinations already present in the training set, while Test Set 2 contained combinations absent from the training set, thus indicating the model's ability to handle novel combinations. The Spearman's correlation coefficients between predicted and observed drug synergy were 0.50 (95% CI: 0.47–0.53) in Test Set 1 and 0.26 (95% CI: 0.22–0.30) in Test Set 2, compared to 0.38 (95% CI: 0.34–0.42) and 0.18 (95% CI: 0.16–0.20), respectively, for the best performing method in the Challenge. We show evidence that higher synergy is associated with higher functional interaction between the drug targets, and this functional interaction information is captured by PAS. We illustrate the use of PAS to provide a potential biological explanation in terms of activated pathways that mediate the synergistic effects of combined drugs. In summary, DIPx can be a useful tool for personalized prediction of drug synergy and exploration of activated pathways related to the effects of combined drugs.

## Introduction

Targeted therapies such as specific inhibitors are the most promising class of cancer drugs, but often fail or achieve only temporary remission due to inherent or acquired resistance. Theoretically, by aiming at multiple targets simultaneously, drug combinations can produce a synergistic interaction that increases drug effectiveness and reduces resistance and the chances of relapse (*Medicine, 2017*; *Pemovska et al., 2018*; *Plana et al., 2022*). This is illustrated in the combination of a BRAF inhibitor dabrafenib with a MEK inhibitor trametinib, which suppresses paradoxical reactivation and resistance observed in patients with BRAF-mutated melanoma treated with dabrafenib alone (*Zhong et al., 2022*; *Banzi et al., 2016*). This recently approved combination has been shown to improve progression-free and overall survival rates (*Subbiah et al., 2023*).

The discovery of effective drug combinations has traditionally relied on expert knowledge and understanding of known biological mechanisms (*Li et al., 2015*). However, this expert-based approach has limited scope to come up with novel combinations. Furthermore, ideally, novel combinations are clinically tested, but it is practically impossible to test all reasonable combinations in a clinical setting. Computational models of drug synergy have shown some potential for personalized prediction of synergistic combinations (*Güvenç Paltun et al., 2021*; *Wu et al., 2022*; *Kong et al., 2022*). These models are typically based on the integration of patient-specific molecular data, such as mutation profiles, gene expression, and drug response information (*Güvenç Paltun et al., 2021*). For example, TAIJI, the best performing method in the AstraZeneca-Sanger (AZS) DREAM Challenge, uses these diverse data types to predict drug synergy (*Li et al., 2018*). The drug combinations predicted to be effective will expand the therapeutic options while maintaining the same level of adverse effects profile. However, despite the advantages offered by modern machine learning methodologies and the availability of large-scale datasets, the prediction of synergistic combinations and validating computational models remains challenging. For example, drug screening protocols often vary across studies, and there is a limited overlap in tested drugs and cell lines, complicating the external validation of these models. Additionally, the reliance on 'black-box' machine learning approaches hinders the exploration of underlying molecular mechanisms driving synergistic combinations.

To address this limitation, several studies have introduced statistical and computational approaches to infer the mechanisms of action of synergistic combinations within cancer signaling pathways. For example, Liu et al. proposed TranSynergy, a drug synergy prediction model that uses the interaction between drug target genes in a protein-protein interaction (PPI) network (*Liu and Xie, 2021*). However, TranSynergy only relies on target gene information, neglecting information on upstream and downstream activities of the targets and their differential contributions to synergy. More recently, Tang et al. developed SynPathy, a deep learning model for drug synergy prediction that incorporates drug-pathway associations (*Tang and Gottlieb, 2022*). SynPathy calculates pathway enrichment scores as a measure of the distance between target genes of each drug in a combination and pathway genes in the PPI network. These pathway enrichment scores, along with chemical structure information, are then combined to fit the model and infer pathway importance scores for each combination. More recently, Wu et al. introduced ForSyn (*Wu et al., 2023*), a deep forest-based method. Although ForSyn implemented a gene enrichment analysis to identify cancer-related pathways, it does not directly identify them through prediction.

Here, we present a Drug synergy Interaction Prediction (DIPx) based on tumor- and drug-specific pathway activation scores (PASs). PASs are biologically motivated features that provide potentially relevant information on the underlying mechanisms of synergistic combinations. We trained and validated DIPx using the AZS DREAM Challenge dataset (*Menden et al., 2019*) and compared its performance with the best performing method in the Challenge. Furthermore, we assessed the generalizability of the model by validating it on the O'Neil dataset (*O'Neil et al., 2016*) and provided illustrations of pathways that could mediate the synergistic combinations found by DIPx. DIPx is publicly available at https://www.github.com/tracquangthinh/DIPx (copy archived at *Trac, 2025*).

## Results

### A pathway-based drug synergy prediction model

*Figure 1* provides an overview of DIPx, which uses gene expression, mutation profiles, gene-interaction network, and drug synergy data to generate PAS of upstream, downstream, and driver genes. Based

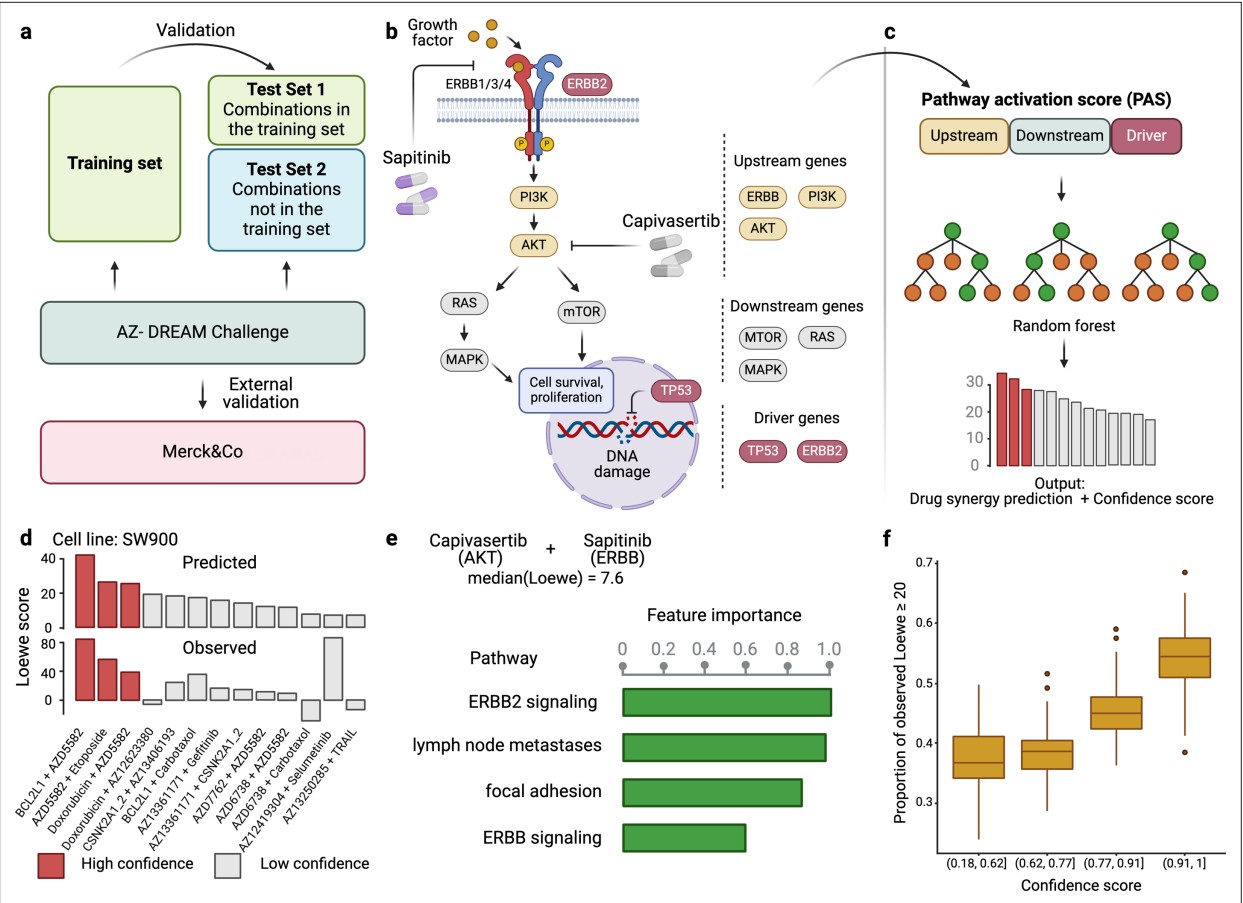

**Figure 1.** Overview of Drug synergy Interaction Prediction (DIPx). (**a**) The AstraZeneca-Sanger (AZS) omics data were used to train and validate the model. The test set was split into two subsets: Test Set 1 contained combinations found in the training set, while Set 2 comprised combinations not found in the training set. The model was also externally validated using the O'Neil dataset. (**b**) A cartoon illustration of the ERBB pathway in a breast cancer cell line treated with the combination of Capivasertib + Sapitinib. Capivasertib targets the AKT gene, whereas Sapitinib targets the ERBB genes. Pathway genes were classified into upstream and downstream genes relative to the position of the target genes in the network. (**c**) The drug synergy prediction model was trained using pathway activation scores (PASs) of the upstream, downstream, and driver genes. (**d**) The predicted and observed Loewe scores of a cell line achieved the median Spearman's correlation in Test Set 1 of the AZS dataset. The color of each bar shows the confidence score information with the threshold of 0.75. (**e**) The main pathways that contribute to the prediction of the synergy of the Capivasertib + Sapitinib combination. (**f**) The proportion of validated high synergistic predictions (Loewe score $\geq 20$) increases with higher confidence scores. The x-axis presents four groups defined by quartiles of confidence scores. This figure was created using BioRender.com.

on the PAS combination of these gene sets, we train a random forest prediction model for DIPx and validate its predictive performance in the AZS dataset; see, e.g., *Ishwaran and Lu, 2019*, for a more detailed description of the random forest model. For a new experiment (drug A+drug B, cell line C), DIPx provides the predicted Loewe score and a confidence score. The test set consisted of two subsets: (i) Test Set 1 includes combinations from the training set, and (ii) Test Set 2 includes combinations absent from the training set. Together, both sets assess the generalizability of the prediction for new patients and new combinations. The analysis involved a total of 75 cell lines tested in 910 combinations in the AZS dataset. DIPx was also validated using an external dataset, as shown in *Figure 1a*.

*Figure 1b* illustrates the ERBB signaling pathway in relation to the Capivasertib + Sapitinib combination, where the genes belonging to the pathway are classified into upstream and downstream genes relative to the position of the target genes: AKT targeted by Capivasertib and ERBB targeted by Sapitinib. Putative driver mutations were identified in each sample based on a well-characterized list of frequently mutated genes in cancer; see section 'Datasets'. We first calculate the PAS of the upstream and downstream part of the pathway relative to the driver genes; see the Materials and methods section for details. The PAS values are then combined to train a random forest regression

model. Given a new drug combination experiment, DIPx predicts the Loewe score for drug synergy, as shown in *Figure 1c*.

*Figure 1d* presents the predicted and observed synergies for the SW900 lung cancer cell line, which has a median correlation of 0.50 among the cell lines in Test Set 1; each bar in the figure represents a drug combination. The best predicted combinations include BCL2L1 + AZD5582, AZD5582 + etoposide, and doxorubicin + AZD5582, with predicted Loewe scores of 42.34, 26.60, and 25.72, respectively, and high confidence scores of 1.0, 0.90, and 0.82, respectively. A combination with a Loewe score greater than 20 is considered highly synergistic (*Menden et al., 2019*). Although the combination of doxorubicin + AZ12623380 is predicted to have high synergy, it is a low-confidence prediction with a confidence score of 0.33. Indeed, the observed Loewe synergy score for this combination is near zero.

The use of PAS allows DIPx to infer the potential biological mechanisms of synergistic drug combinations. *Figure 1e* shows pathways with the highest contribution to the prediction of drug synergy of the Capivasertib + Sapitinib combination: these include the ERBB-related pathways (ERBB2 signaling pathway, ERBB pathway) and tumor-related pathways (lymph-node metastases, focal adhesion).

*Figure 1f* demonstrates the association between the confidence scores and the validation of predictions. The *x*-axis represents the confidence scores grouped into quartiles, while the *y*-axis displays the proportion of validated high synergy (Loewe score $\geq$ 20). Predictions with higher confidence scores are expected to exhibit a greater level of validation. Indeed, in this figure, the proportion of high synergistic predictions that are validated in the combination of Test Set 1 and Test Set 2 of the AZS dataset increases as the confidence score rises.

## Validation and comparisons in the AZS dataset

We evaluated the performance of DIPx in the AZS test sets and compared it with TAIJI, which was the best performing method in the AZS DREAM Challenge (*Li et al., 2018*). TAIJI was trained using both monotherapy drug response and molecular data. Since DIPx uses only molecular data, to make a fair comparison, we trained TAIJI using only molecular features and referred to it as TAIJI-M.

*Figure 2a* shows the correlation between the predicted and observed Loewe scores of 963 experiments in Test Set 1 ($r = 0.5$, 95% CI: 0.47–0.53), where each experiment represents a combination of drug A+drug B tried on cell line C, yielding one data point. In comparison, TAIJI-M gives $r = 0.38$ (95% CI: 0.34–0.42). We also bootstrapped the training set ($n$=100 times), and for each bootstrap replicate, calculated the Spearman's correlation between the predicted and observed scores of all experiments. As illustrated in *Figure 2b*, DIPx achieved stable Spearman's correlations across all bootstrap replicates, which are significantly higher than those of TAIJI-M. The bootstrap distribution actually indicates that the Spearman's correlation from DIPx is negatively biased, while from TAIJI it is slightly positively biased. This means that the gap between the bias-corrected estimates of the Spearman's correlations from DIPx and TAIJI-M would be even larger; see the Materials and methods section for a theoretical explanation.

To demonstrate that DIPx does not overfit the training set, we performed a 10-fold cross-validation for DIPx. *Figure 2—figure supplement 1* shows the Spearman's correlation between the predicted and observed Loewe scores across the 10 folds. DIPx achieved a median correlation of 0.48, which is comparable to the correlation of 0.50 in Test Set 1. This indicates that there is no evidence of overfitting in the training set.

Furthermore, we compared the performance of DIPx and TAIJI-M across all cell lines in Test Set 1 using a Spearman's correlation between the predicted and observed synergy scores, as shown in *Figure 2c*. A majority of the cell lines (63%) are below the diagonal line, indicating that DIPx outperforms TAIJI-M in predicting synergy scores for these cell lines.

We also compared the performance of DIPx and TAIJI-M in Test Set 2. As expected, the prediction performance of both methods was worse in Test Set 2 than in Test Set 1 since Test Set 2 consists of new combinations absent from the training set. The Spearman's correlation of the observed vs predicted synergy using DIPx is 0.26 (95% CI: 0.22–0.30), which is greater than 0.18 (95% CI: 0.16–0.20) using TAIJI-M. However, the difference is not statistically significant. *Figure 2d* shows that this result is stable across 100 bootstrap replications of the training set. A similar downward bias for DIPx is observed in the bootstrap distribution.

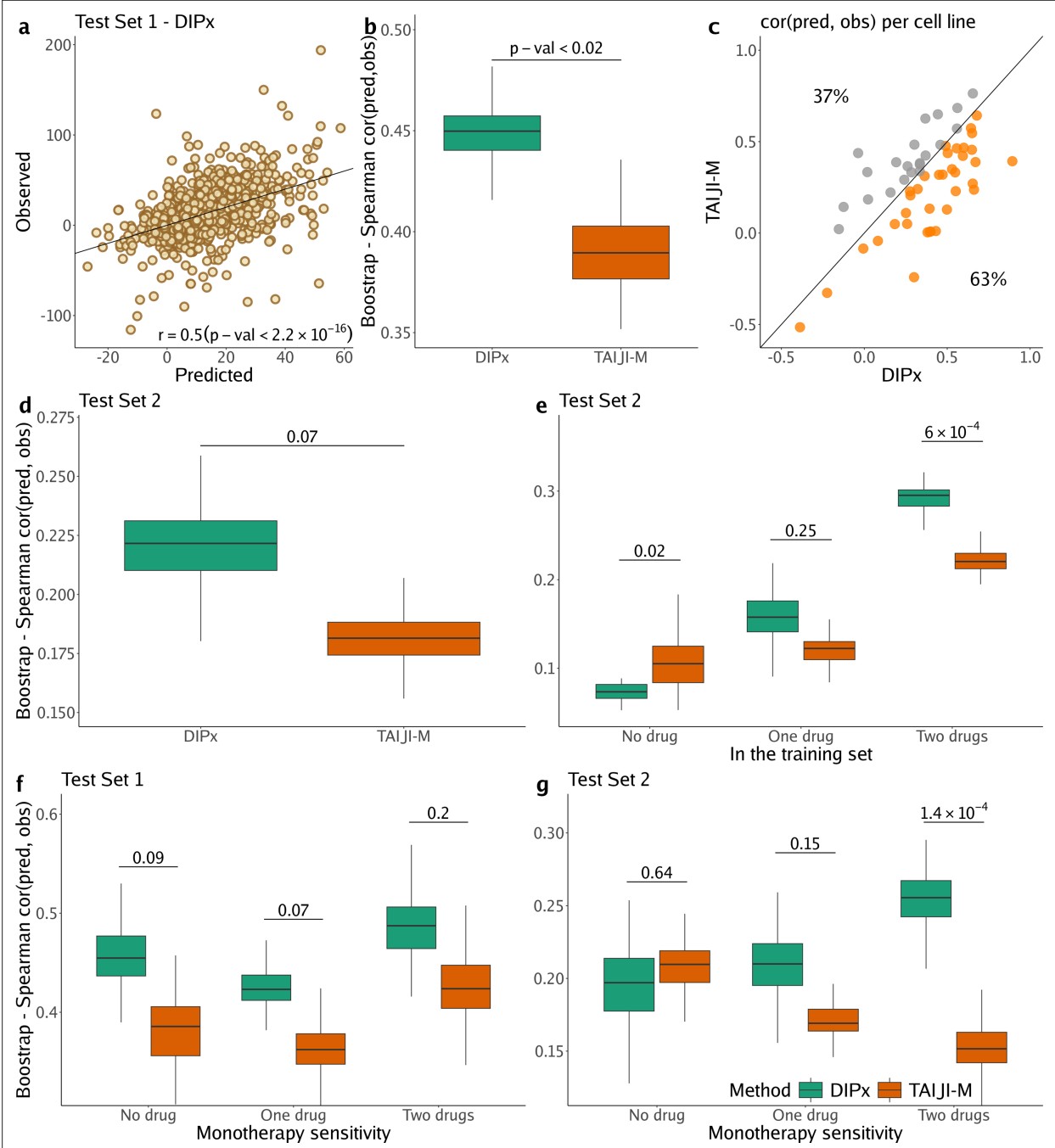

**Figure 2.** Performance of Drug synergy Interaction Prediction (DIPx) in the AstraZeneca-Sanger (AZS) dataset. This includes both Test Set 1 (panels a, **b, c, f**) and Test Set 2 (panels **d, e, g**). (**a**) Comparison of predicted vs observed synergy scores for all experiments in Test Set 1. (**b**) Comparison of DIPx vs TAIJI-M in terms of the correlation between predicted and observed synergy scores from all experiments in Test Set 1. Each box plot shows the results based on 100 bootstrap replicates of the training set. (**c**) Comparison of DIPx and TAIJI-M performance across cell lines in Test Set 1. Each point represents the correlation between the predicted and observed synergy for a given cell line. (**d**) Comparison of DIPx vs TAIJI-M in Test Set 2. Each box plot displays the correlations between the predicted and observed values obtained from 100 bootstrap replicates of the training set. (**e**) Comparison of performance between DIPx and TAIJI-M in Test Set 2 in relation to the number of drugs in common (x-axis) between the combinations in the test set and the training set. (**f** and g) DIPx vs TAIJI-M in three groups classified by monotherapy sensitivity of two drugs in a combination in Test Set 1 (**f**) and Test Set 2 (**g**).

The online version of this article includes the following figure supplement(s) for figure 2:

**Figure supplement 1.** 10-fold cross-validation of Drug synergy Interaction Prediction (DIPx) in the training set of the ASZ DREAM dataset.

To investigate the effect of unseen combinations on prediction performance, we divided each combination (drug A+drug B) in Test Set 2 into one of three groups based on the number of individual drugs present in the training set: (i) neither drug A nor drug B in the training set ('no drug'), (ii) either drug A or drug B in the training set ('one drug'), (iii) and both drugs A and B in the training set ('two drugs'), as shown in *Figure 2e*. Overall, both DIPx and TAIJI-M showed improved performance as the number of drugs present in the training set increased. For experiments in which both drugs were not in the training set ($n = 262$), TAIJI-M achieved a median correlation of 0.11, while DIPx performed worse with a median correlation of –0.03. For experiments with at least one drug in the training set ($n = 2499$), both methods showed improved performance with median correlations of 0.16 and 0.12 for DIPx and TAIJI-M, respectively. When both drugs in an experiment were present in the training set ($n = 4370$), DIPx achieved a median correlation of 0.30, which was better than TAIJI-M's performance ($r = 0.22$, p-value $< 6 \times 10^{-4}$).

Monotherapy drug response profiles have been shown to correlate with synergistic effects and contribute to improving prediction performance, e.g., in TAIJI (*Li et al., 2018*). Here, we compared the performance of DIPx and TAIJI-M in relation to monotherapy sensitivity as measured by the IC50 value. We categorized each experiment in the AZS test sets into three groups according to the monotherapy response. Briefly, we first calculated the median sensitivity to monotherapy for each drug A ($T_A$) across all experiments. Measuring the response of a cell line to drug A in an experiment by $S_A$, the drug is considered sensitive if $S_A \geq T_A$. We then compared the synergy of a combination of drugs A and B in relation to the monotherapy sensitivity to both drugs, only one drug, or neither drug.

In Test Set 1, we observe some improvement by DIPx in all three groups of monotherapy sensitivity, with the highest performance in the group sensitive to both drugs (median $r = 0.48$), but they are not statistically significant, see *Figure 2f*. In Test Set 2, TAIJI-M and DIPx performed similarly in the group with no sensitive drug (median $r = 0.21$ vs $r = 0.20$ by DIPx, p-value $\sim 0.68$). Interestingly, we found

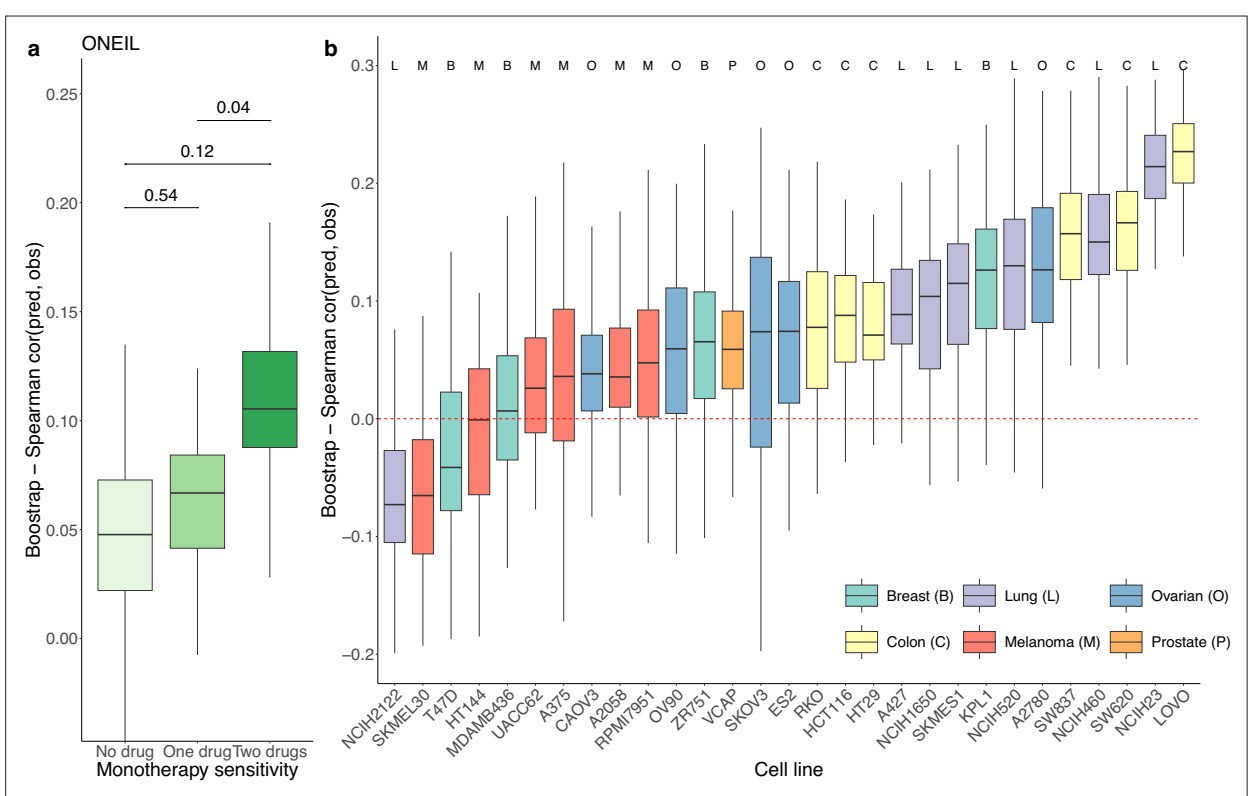

**Figure 3.** Prediction performance of Drug synergy Interaction Prediction (DIPx) in the O'Neil dataset. (**a**) Monotherapy sensitivity, (**b**) 29 cell lines from six cancer tissues. The y-axis in all box plots shows the Spearman's correlation between predicted and observed values in 100 bootstrap replicates.

The online version of this article includes the following figure supplement(s) for figure 3:

**Figure supplement 1.** Prediction performance of Drug synergy Interaction Prediction (DIPx) in the O'Neil dataset, grouped by unseen combinations in the training set (x-axis).

that, while the performance of DIPx improved as the number of monotherapy-sensitive drugs in a combination increased, the performance of TAIJI-M decreased, see *Figure 2g*. All prediction results are provided in *Supplementary file 1A*.

## External validation of DIPx in the O'Neil dataset

We used a similar computational approach to evaluate the prediction performance of DIPx in relation to the sensitivity of the constituent monotherapies and the impact of unseen combinations in the O'Neil dataset. As shown in *Figure 3a*, the performance of DIPx improved with an increasing number of monotherapy-sensitive drugs in the combination, consistent with the results of Test Set 2 of the AZS data. The highest Spearman's correlation between the predicted and observed scores was seen in combinations with two sensitive drugs (median $r = 0.11$). In relation to the number of drugs in a combination present in the training set, DIPx achieved better performance for combinations with none or one drug in the training set (middle box plot, *Figure 3—figure supplement 1*).

We also obtained TAIJI-M's results in the O'Neil dataset. The original version of TAIJI-M uses gene expression, CNV, mutation, and methylation data. However, due to the lack of methylation data in the O'Neil dataset, we retrained TAIJI-M by excluding the methylation features. According to the final report of TAIJI in the challenge (https://www.synapse.org/Synapse:syn5614689/wiki/396206), Guan et al. reported that methylation features do not contribute to prediction performance in the post-challenge analysis. This means that retraining TAIJI-M without methylation data will not affect the comparison between DIPx and TAIJI-M on the O'Neil dataset.

TAIJI-M relies on a gene-gene interaction network to calculate posttreatment gene expression. This approach limits its applicability to new datasets, as TAIJI-M can only predict synergy scores for drug combinations present in the training dataset. Among the set of drug combinations with both drugs present in the training set, both DIPx and TAIJI-M perform poorly, with Spearman's correlations between predicted and observed synergy scores of 0.09 and 0.05, respectively. The poor performance could be due to the limited number of drug combinations (42/583).

We also analyzed the prediction performance of DIPx in the 29 cell lines from six different cancer tissues (*Figure 3b*). Colon cancer (yellow box plots) and lung cancer cell lines (purple box plots) showed better validation compared to cell lines from breast, ovarian, melanoma, and prostate cancers.

## Inference of the mechanism of action based on PAS

The use of PAS in DIPx allows us to infer the potential mechanisms of action of drug combinations while maintaining the prediction performance of the model. For instance, in Test Set 1 of the AZS data, DIPx suggests the involvement of ERBB2 signaling pathways in the Capivasertib + Sapitinib combination, as illustrated by the top pathways depicted in *Figure 1e* and marked yellow in *Figure 4a*. This combination therapy has shown promise in overcoming resistance to anti-ERBB2 monotherapy in HER2+ breast cancer (*Fujimoto et al., 2020*), and ERBB2 has been identified as a key biomarker associated with synergistic responses to this combination in the AZS DREAM Challenge study (*Menden et al., 2019*).

*Figure 4a* further shows the distribution of feature importance vs PAS for all pathways for Capivasertib + Sapitinib combination. The feature importance value (*x*-axis) is calculated using the permutation method of *Ishwaran and Lu, 2019*. The PAS value (*y*-axis) represents the median PAS across samples treated with this combination in two test sets. Our focus is on pathways with high feature importance (e.g. the top 5%) as well as highly activated (top 5% PAS). Therefore, the top-right section of *Figure 4a* is the interesting region. We present additional examples to further demonstrate the capabilities of DIPx. *Figure 4b* gives the top pathways of MEDI3622, an ADAM17 inhibitor, in combinations with AKT inhibitors including Capivasertib and MK-2206. These ADAM17 + AKT combinations target multiple parts of the PI3K/AKT pathway through ERBB activation (*Menden et al., 2019*), which aligns with the potential pathway candidates suggested by DIPx.

One of the key strengths of DIPx is its ability to infer potential mechanisms of both known and novel drug combinations, even in cases where limited biological or clinical information is available. This capability is particularly valuable for new combinations that have not been included in the training set. For instance, in *Figure 4c*, we present the key pathways identified for the Selumetinib + MK-2206 combination from Test Set 2 of the AZS data. We observe the involvement of RAS signaling, with Selumetinib targeting MEK and MK-2206 targeting AKT, as shown in *Figure 4—figure supplement*

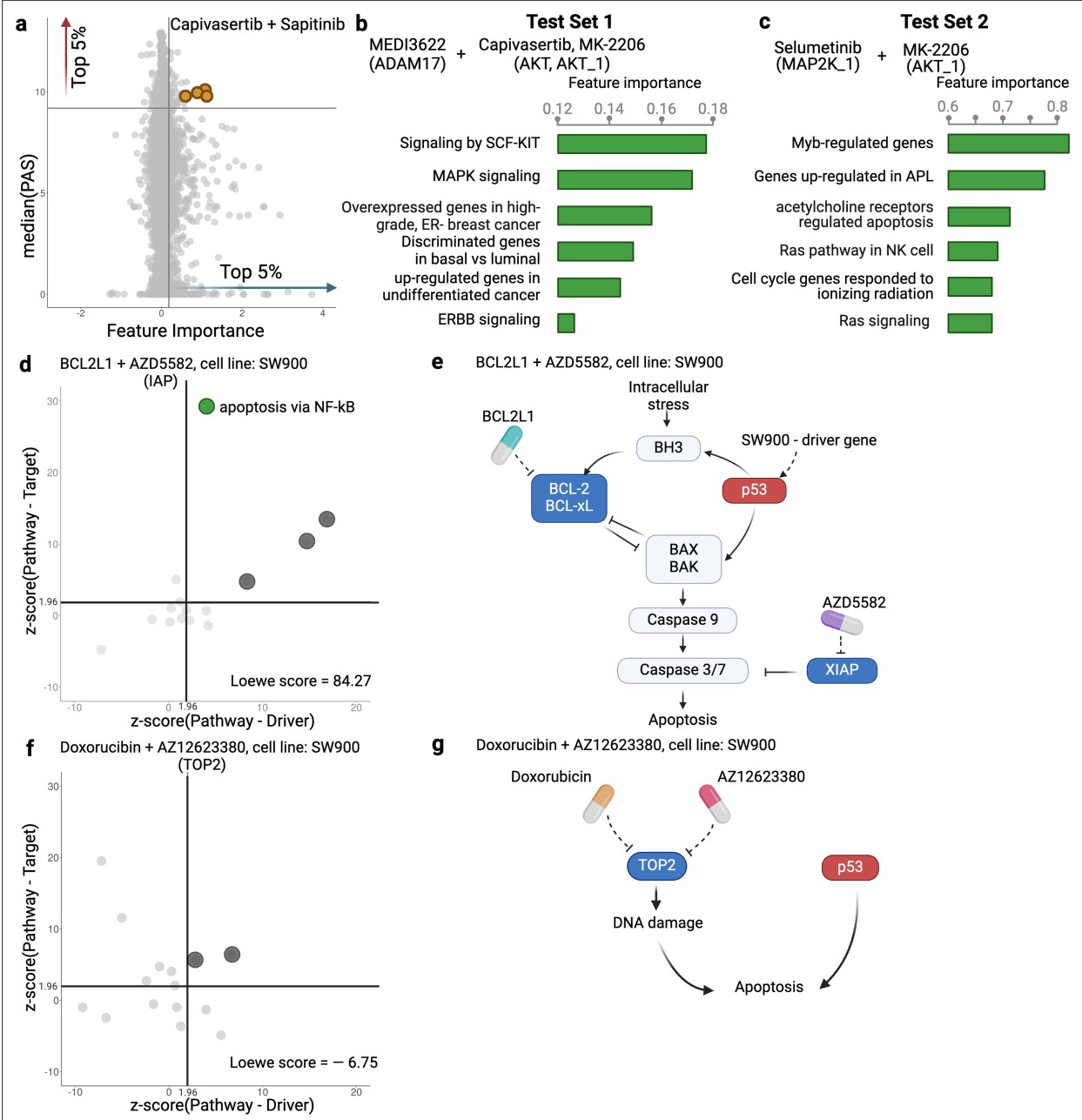

**Figure 4.** Inference of pathway importance scores in the AstraZeneca-Sanger (AZS) dataset. (**a**) Scatter plot showing feature importance (*x*-axis) vs pathway activation score (PAS) (*y*-axis) for the Capivasertib + Sapitinib combination. Pathways with high PAS and feature importance (top 5%) are of particular interest. (**b, c**) Top pathways contributing to the prediction of the combinations in Test Set 1 (**b**) and Test Set 2 (**c**). For each pathway, the bar plots show its feature importance. (**d, f**) Functional interaction between the pathway vs driver genes (*x*-axis) and the pathway vs target genes (*y*-axis) of the top pathways suggested by Drug synergy Interaction Prediction (DIPx) in the SW900 cell line treated with synergistic combination BCL2L1 + AZD5582 (**d**) and the non-synergistic combination Doxorubicin + AZ12623380 (**f**). The *z*-score from network enrichment analysis (NEA) is a measure of functional interaction between two gene sets. A higher *z*-score indicates a stronger interaction compared to a random permutation of the network. The upper right quadrant (*z*-score > 1.96) represents pathways that are potentially interesting. (**e, g**) Cartoon illustration of the potential pathways mediated by the synergistic combination of BCL2L1 + AZD5582 (**e**) and the non-synergistic combination Doxorubicin + AZ12623380 (**g**).This figure was created using BioRender.com.

The online version of this article includes the following figure supplement(s) for figure 4:

**Figure supplement 1.** A cartoon illustration of the RAS pathway mediated by the Selumetinib + MK-2206 combination.

**Figure supplement 2.** Observed (red lines) vs predicted inhibition (black, dash lines) from the Loewe reference model in the SW900 cell line treated by the synergistic BCL2L1 + AZD5582 combination (**a**) and the non-synergistic Doxorubicin + AZ12623380 combination (**b**).

*Figure 4 continued on next page*

*Figure 4 continued*

**Figure supplement 3.** Functional interaction (*x*-axis) between the pathway vs driver genes (first column), the pathway vs all target genes (second), the pathway vs BCL2L1 target genes (third), and the pathway vs AZD5582 target genes (fourth) of the top pathways suggested by Drug synergy Interaction Prediction (DIPx) in the SW900 cell line treated with synergistic combination BCL2L1 + AZD5582.

**Figure supplement 4.** Functional interaction (*x*-axis) between the pathway vs driver genes (first column), the pathway vs all target genes (second), the pathway vs Doxorubicin target genes (third), and the pathway vs AZ12623380 target genes (fourth) of the top pathways suggested by Drug synergy Interaction Prediction (DIPx) in the SW900 cell line treated with non-synergistic combination Doxorubicin + AZ12623380.

**Figure supplement 5.** Top pathways contributing to the prediction of the MK2206 + Erlotinib combination in the O'Neil dataset.

*1*. A recent clinical study has used Selumetinib + MK-2206 to target downstream components of the RAS pathway (*Chung et al., 2017*).

If the drugs in a combination have the same target, the efficacy of the combination is likely similar to that of each individual drug at higher doses, i.e., they will only have an additive effect. So it seems reasonable to hypothesize that a synergistic combination is more likely to occur when the two drugs have different targets (*Chen et al., 2015*). But how should the targets be related to each other? To investigate this, we examine the pathways suggested by DIPx. First, we choose a synergistic combination of BCL2L1 + AZD5582 in the SW900 cell line for further illustration. The contour plot of the BCL2L1 + AZD5582 inhibition in the SW900 cell line is illustrated in *Figure 4—figure supplement 2a*. We first collected the top 15 pathways (ranked by feature importance) for this BCL2L1 + AZD5582 combination suggested by DIPx. The full list of these pathways is shown in *Figure 4—figure supplement 3*. *Figure 4d* illustrates the functional interaction between the genes of the top 15 pathways and the driver genes of the SW900 cell line (*x*-axis), alongside the target genes of the combination BCL2L1 + AZD5582 (*y*-axis). To assess the strength of this interaction, we used the network enrichment analysis (NEA) (*Alexeyenko et al., 2012*), which provides *z*-score, an enrichment score, indicating the degree of interaction. A higher *z*-score reflects a stronger interaction between the two gene sets. The top pathways exhibiting high functional interaction with both the driver genes and target genes (*z*-score > 1.96) are particularly notable, located in the upper right quadrant of *Figure 4d*. In particular, the apoptosis pathway via NF-kB (highlighted in green) has the highest pathway-target interaction among these pathways. *Figure 4e* shows the cartoon illustration of the pathway in which the drug BCL2L1 targets BCL-xL and AZD5582 targets XIAP. This suggests an explanation for the observed synergy between the two drugs. Thus, it appears that in this case we get synergy when the drugs target different parts of a driving pathway, either directly or via other functional interactions.

As a negative control, we examine the non-synergistic combination Doxorubicin + AZ12623380, which targets the same gene TOP2; see *Figure 4f and g*, *Figure 4—figure supplement 2b*. We similarly obtain 15 top-ranking pathways according to feature importance, but now we do not expect to see anything obviously relevant to the SW900 cell line (more details in *Figure 4—figure supplement 4*). Some pathways that have a high functional interaction with the target genes (upper-left quadrant) have little interaction with the drivers. There are no clearly outlying points in the upper-right quadrant; the two pathways near the boundary are (i) Shen_Smarca2_targets_up, containing genes whose expression was negatively correlated with the expression of the SMARCA2 gene in prostate cancer samples, discovered in relation to androgen-induced proliferation in the prostate; and (ii) Kokkinakis_Methione_deprivation_48hr_up, which contains upregulated genes in melanoma cell-line MEWO cells after 48 hr of methionine deprivation. They do not appear to be relevant for the lung cancer cell line SW900.

We also applied DIPx to identify potential activated pathways in the O'Neil dataset. *Figure 4—figure supplement 5* highlights the key pathways contributing to the MK2206 + Erlotinib combination. The most significant pathway is 'Metabolism by CYP Enzymes'. Previous studies *Molife et al., 2014* have reported that both MK2206 and Erlotinib are metabolized by the CYP enzyme family, further supporting this finding.

## PAS captures the functional interaction of drug targets

In *Figure 5a*, using the AZS data, we compare the observed drug synergy of combinations of two drugs that share some target genes vs those that do not share any target genes. No significant differences were observed (p-value>0.72), suggesting that nonoverlapping drugs in terms of their targets do not necessarily result in improved drug synergy.

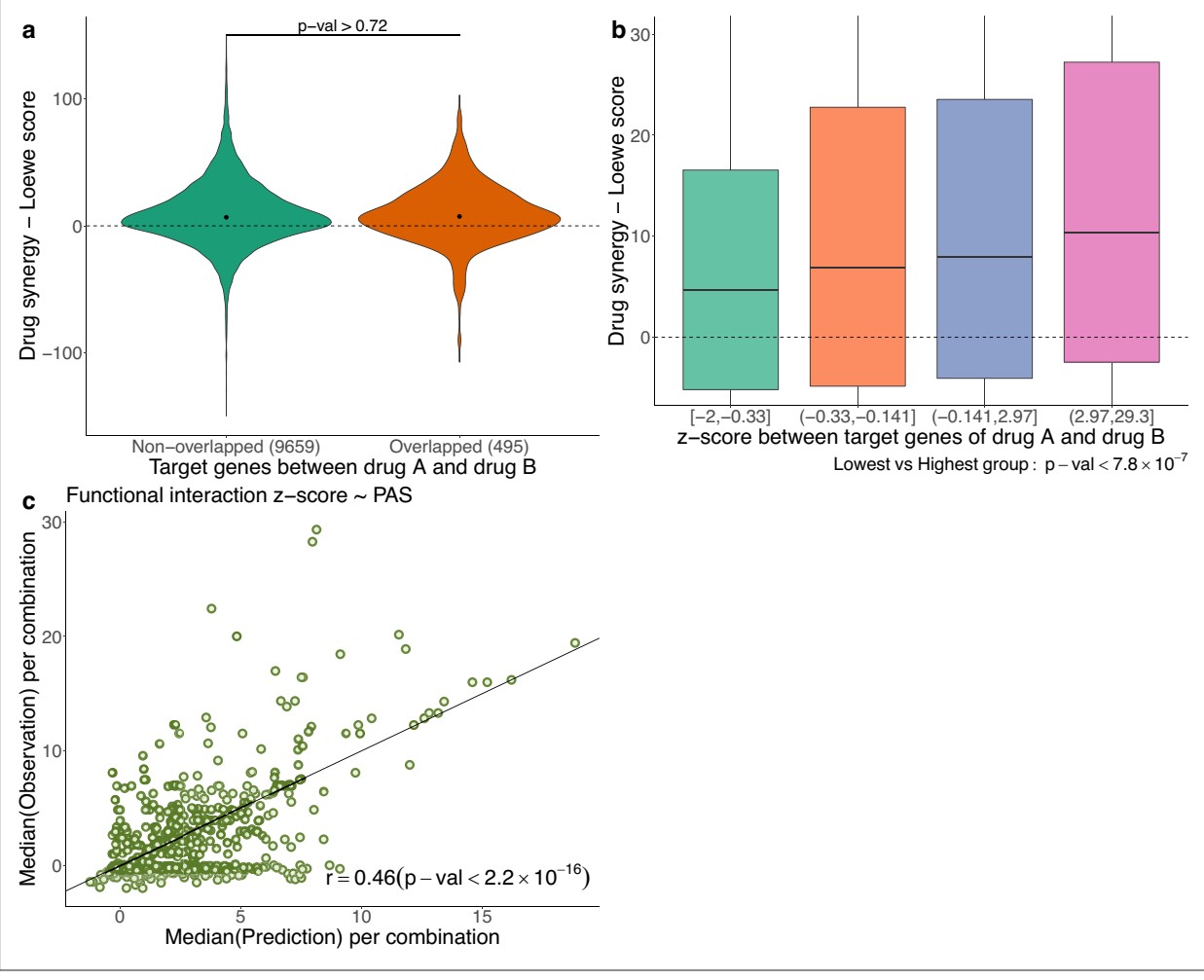

**Figure 5.** Functional interaction of drug targets in drug synergy and PAS. (**a**) Comparison of drug synergy between combinations (drug A+drug B) with vs without overlapping target genes. The numbers in parentheses show the sample sizes of each group. (**b**) Drug synergy between four groups in relation to increased functional interaction between the target genes of the two drugs. (**c**) Comparison between the observed functional interaction (z-score in the network enrichment analysis) and the predicted z-score by pathway activation score (PAS).

However, we also observed synergy when the two drugs target different genes in the same pathway. More generally, we hypothesize that synergistic effects occur when the targets have functional interaction. As before, the functional interaction is assessed using NEA (*Alexeyenko et al., 2012*), where a higher z-score value indicates a stronger functional interaction between the two drugs. *Figure 5b* shows the observed drug synergy (y-axis) in the AZS data for the four groups defined by the quartile values of the z-scores (x-axis). It indicates that combinations with higher functional interaction are more likely to achieve higher drug synergy, with the highest z-score group ($z \in (2.97, 29.3)$), exhibiting the most favorable drug synergy (median Loewe score = 10.34).

However, when added to the prediction model, the functional interaction z-score did not improve the prediction of synergy (data not shown). Statistically, this can happen if PAS already captures the functional interaction information. To show this, using the AZS training data, we trained a prediction model using PAS as the feature and the functional interaction z-score as the output. We then evaluated the performance of the model in the test set. As shown in *Figure 5c*, we observed a significant correlation between the predicted and observed z-scores, with a Spearman's correlation coefficient of 0.46. This explains why the functional interaction does not give additional predictive power in our model. All medians of predicted and observed Loewe scores related to *Figure 5c* are provided in *Supplementary file 1B*.

## Discussion

We have developed and validated DIPx, an advanced computational model that incorporates gene expression and mutation profiles to predict synergistic drug combinations. DIPx performs well against the best performing method in the AZS DREAM Challenge. Through the use of tumor- and patient-specific PASs, DIPx also provides valuable information on the potential underlying pathways associated with an observed synergistic drug interaction. In addition to rigorous validation using the AZS dataset, DIPx is further validated on the independent O'Neil dataset. This comprehensive validation ensures the robustness and reliability of DIPx in predicting drug synergy across different cancers.

We have compared the performance of DIPx to TAIJI-M, the molecular-based model of TAIJI (*Li et al., 2018*). The extra information from the use of monotherapy data in TAIJI is rather small, approximately a 10% increase in the overall Spearman's correlation (*Li et al., 2018*), and, of course, we could also use such data in DIPx, but it is more convenient and informative to focus the comparisons on prediction based on molecular data alone. For instance, this allows us to compare DIPx with TAIJI-M on the prediction of combinations that contain untrained drug(s), which is not possible with the full TAIJI.

The recent availability of large-scale drug combination assay data has allowed the development of realistic prediction models for drug synergy. These datasets offer a substantial number of samples encompassing hundreds of combinations, allowing for extensive validation studies. However, it is important to note that these datasets were generated using different protocols and drug screening techniques. For instance, the AZS data used a 5-by-5 concentration matrix, while the study by O'Neil et al. used a 4-by-4 format. In addition, there is limited overlap in the cell lines used among the datasets. These differences pose challenges to the proper validation of prediction methods (*Menden et al., 2019*). Exploring new datasets or applying novel techniques in the training process (e.g. transfer learning) will be our future direction to improve the performance of DIPx.

A particular strength of our study is that we use the best performing method in the Challenge as a benchmark. This is convenient and robust benchmarking, as there were 160 teams that participated in the Challenge (73 teams submitted in the final round). Altogether, these teams used practically all of the commonly used machine learning tools; see the summary in *Menden et al., 2019*. Another strength is our use and validation of the confidence score metric, which captures the statistical uncertainty in the predicted synergy by a single number. This is more convenient for clinical interpretation than the standard prediction interval, because there is a target level for which a combination is considered synergistic, so the score measures our confidence in achieving the target.

Despite promising results, our study has several limitations. First, the use of cell lines as training and validation samples from the AZS and O'Neil datasets may not fully capture the heterogeneity present in actual tumors. Second, the computation of PAS relies solely on the primary target genes of the drug combinations, potentially disregarding valuable information from non-primary targets. There could also be off-targets that we do not know about. This limitation might lead to the loss of information about the broader effects of drug combinations. Third, cancer is a heterogeneous disease that occurs in many tissues. Even within a single tissue, cancer exhibits distinct (molecular) subtypes with varying biological mechanisms and clinical outcomes. Since DIPx was developed using pan-cancer datasets, it may not be optimal for tissue-specific predictions. Our future plan for DIPx would be building cancer-specific models.

Last but not least, the prediction of previously untrained combinations remains a great challenge. The worst case is for combinations of drugs that were not previously trained, with the Spearman's correlation only around 0.1. However, from a clinical perspective, it is perhaps more realistic to look for combinations among drugs previously trained in monotherapy or in other combinations. Improving the prediction for the combination of such drugs would be worthwhile.

## Materials and methods
### PAS for drug combinations

PASs are the key features in DIPx. The PAS of pathway P in cell line C is calculated for each drug combination (drug A+drug B) and pathway P. Genes in pathway P are grouped into three subgroups: (i) $G_u$, which includes all the target genes of drugs A and B, as well as the upstream genes of pathway P; (ii) $G_d$, which includes the downstream genes of pathway P; and (iii) $G_{dr}$, which consists of all the

driver genes of cell line C in pathway P. In the example of the ERBB pathway targeted by Capivasertib + Sapitinib (**Figure 1b**), $G_u$ consists of ERBB, PI3K, and also AKT; $G_d$ contains MTOR, RAS, and MAPK, while $G_{dr}$ includes TP53 and ERBB2.

The score for upstream activity ($PAS_u$) is calculated by the sum of mRNA expression for genes in $G_u$. Similarly, the scores for the downstream activity ($PAS_d$) and the set of driver genes ($PAS_{dr}$) are calculated from $G_d$ and $G_{dr}$. In practice, the genes of the N=4762 curated human pathways are provided from the MsigDB database (version 6.2) (**Liberzon et al., 2015**). The target genes of the drugs are collected from the AZS dataset and extended from the DrugBank database (**Wishart et al., 2018**) and the ChEMBL database (**Zdrazil et al., 2024**). The extraction of the driver genes of the cell lines is described in section 'Datasets'.

## A pathway-based model for drug synergy prediction

The training features of DIPx consist of three components: upstream activity ($PAS_u$), downstream activity ($PAS_d$), and driver genes ($PAS_{dr}$), as shown in **Figure 1b**. The final training matrix has a size of K experiments by 14286 PASs, where each row corresponds to a specific experiment (drug A+drug B, cell line C).

To address potential sparsity in the training matrix caused by pathways with no target or driver genes, we explored an alternative model with N=4762 additional features. Each feature corresponds to a pathway P and is calculated as $S(g) * (w1 + w2)$, where $S(g)$ represents the sum of mRNA expression for all genes in pathway P, and w1 and w2 denote the functional interactions between gene sets: (pathway genes ↔ target genes) and (pathway genes ↔ driver genes), respectively. The functional interactions were estimated using NEA and converted into normal probability scores for w1 and w2. The feature value is zero only when the pathway lacks both targets and driver genes, as well as any interactions with drug targets and driver genes. Additionally, we incorporated the NEA enrichment score between target genes and driver genes into the final matrix. Despite adding these new features, the alternative model did not exhibit any significant improvements in predictive power (data not shown).

For the predictor, we used the random forest algorithm implemented in the randomforestRSC package (with default parameters) in R version 4.0.4. During the development of DIPx, we experimented with various machine learning methods, such as the support vector machine and the elastic net. However, we found that these other methods yielded comparable results and that tuning their parameters did not significantly improve prediction performance while requiring extensive additional computations (data not shown). The random forest algorithm in the randomforestRSC package also offers multiple options to calculate the importance of features. In this study, we used the permutation (or Breiman-Cutler) method (**Ishwaran and Lu, 2019**) to infer the importance of each PAS.

The confidence score (CS) is used to assess the statistical quality of synergy prediction; see Section 5.6 in **Pawitan, 2001**, for the confidence concept in general. First, as previously defined for example in **Menden et al., 2019**, a combination is considered synergistic if the Loewe score is greater than or equal to 20. For each sample $s$, we have the actual predicted synergy $P_s$ (the output of the regression random forest model). We then generate $N_b = 100$ bootstrap replicates of the training data and obtain the bootstrap predictions for the sample: $P_{s1}^*, \ldots, P_{s100}^*$. The CS of $P_s$ is defined as follows:

$$\mathrm{CS}(Ps) = \frac{\#(P_{si}^*s \geq 20)}{N_b}.$$

The bootstrap replicates are also used to evaluate the standard errors (se) of the Spearman's correlation between the observed and predicted synergy scores in the test sets. The 95% confidence intervals are computed by the usual formula: $\widehat{\rho} \pm 1.96\mathrm{se}$, where $\widehat{\rho}$ is the observed Spearman's correlation. Though less frequently used, the bootstrap can also be used for bias correction (**Pawitan, 2001**, Section 5.2). Bias occurs if there is a nontrivial gap between the observed estimate and the mean of the bootstrap replications, and bias correction is used to adjust the original estimate. Theoretically,

$$\mathrm{Bias} = E_F(\widehat{\rho}) - \rho,$$

where F is the underlying data distribution. So, the bias-corrected estimate should be

$$\widehat{\widehat{\rho}} = \widehat{\rho} - \mathrm{Bias}$$

In practice, the bias is estimated by

$$\widehat{\text{Bias}} = E_{\widehat{F}}(\widehat{\rho}) - \widehat{\rho}$$
$$= \text{average}\left\{\widehat{\rho}_1^* \cdots \widehat{\rho}_n^*\right\} - \widehat{\rho},$$

where $\widehat{\rho}_1^* \ldots \widehat{\rho}_n^*$ are the bootstrap replicates of $\widehat{\rho}$. When the estimated bias is negative, as we observed for DIPx, the bias-corrected estimate is shifted upward. And vice versa, if the bias is positive, as observed for TAIJI-M, the corrected estimate is shifted downward.

## Computing p-values using the bootstrap

To compare the predictive performance of DIPx and TAIJI-M (e.g. as shown in *Figure 2*), the bootstrap method can be used to compute a confidence interval for differential correlation in the test set. However, there is a close relationship between p-values and confidence intervals; see *Pawitan, 2001*, Chapter 5; particularly p.134. In this case, we compute the p-value as follows:

(1) For each bootstrap replication, (i) compute the Spearman's correlation between the predicted and observed scores in the test set for DIPx and TAIJI-M. Denote this by $r_1$ and $r_2$. (ii) Compute the difference in the Spearman's correlations $d = (r_1 - r_2)$.

(2) Repeat the bootstrap $n = 100$ times.

(3) Compute the minimum of these two proportions: proportion of $d<0$ or proportion of $d>0$. To overcome the limited bootstrap sample size, we use the normal approximation in computing the proportions.

(4) The two-sided p-value = 2 × the minimum proportion in (3).

## Datasets

### AZS DREAM Challenge dataset

The AZS DREAM Challenge is a rigorous competition in the effort to systematically develop and validate drug synergy prediction methods. Indicating the strong interest in the topic, 160 international teams (*Menden et al., 2019*) participated in the Challenge. It was organized into two subchallenges: (i) Prediction for known (tested) combination and (ii) Prediction for unknown (untested) drug combinations. The final dataset comprised 11,576 experiments from 85 cell lines and 910 combinations. The gene expression data of these cell lines was obtained from Affymetrix microarray (*Menden et al., 2019*). However, to ensure consistency between the AZS dataset and the O'Neil dataset (*O'Neil et al., 2016*) (which did not provide gene expression profiles of cell lines), we utilized gene expression data from the Cancer Cell Line Encyclopedia (CCLE) cohort (*Ghandi et al., 2019*).

Out of the 85 cell lines, we identified 75 cell lines with available gene expression data in the CCLE cohort, resulting in a total of 10,154 experiments involving 903 combinations used in our study. *Supplementary file 1C* shows the list of 75 cell lines. For the validation of the prediction model, the data were split into a training set (n=2060) and two test sets (n=963 and 7,131) according to subchallenges 1 and 2, respectively. The first test set contains experiments from 167 combinations (of 69 single drugs) that are also in the training set. The second test set includes experiments with 736 drug combinations that are not in the training set.

We collected gene expression data of 75 cell lines, measuring the transcripts per million of 37222 genes, of the CCLE cohort downloaded from the DepMap Portal (*Tsherniak et al., 2017*). The gene expression data was logarithmically transformed to the base 2 scale for downstream analysis. Additionally, we obtained potential driver genes for these cell lines, including both mutations and fusion genes, from the DepMap Portal. The portal provides information on mutations in 1637 protein-coding genes associated with cancer biology in a collection of 1030 cell lines.

To filter the list of mutations, we focused on those occurring in at least 2.5% of the total cell lines. Subsequently, we extracted the list of mutations specific to the 75 cell lines under investigation. For fusion genes, we focused on those present in the Miltelman database (*Mitelman, 2022*) and occurring at least twice, considering them as relevant for our analysis. The final list of potential driver genes for the 75 cell lines can be found in *Supplementary file 1C*. On average, each cell line had a median of 29 potential driver genes.

For the drug synergy data, we used a 5-by-5 concentration matrix provided by the Challenge. Drug synergy values were estimated using the Loewe reference model from Combenefit (*Di Veroli et al., 2016*).

### O'Neil dataset

O'Neil dataset is a large-scale drug synergy screening dataset from Merck&Co company (*O'Neil et al., 2016*). A total of 23062 experiments with 583 unique drug combinations (38 monotherapy drugs) were carried out on 38 cancer cell lines by a 4-by-4 drug concentration matrix. Out of 38 cell lines, we found 29 cell lines with available gene expression data from the DepMap Portal. The details of the 29 cell lines are described in *Supplementary file 1D*. The gene expression data of 37,222 genes from 29 cell lines, as well as the driver genes of these cell lines, were collected from the DepMap Portal using the same procedure as in the AZS dataset. The original release of this dataset provides only the raw data on drug synergy. Here, we calculated the Loewe synergy score for each experiment using Combenefit (*Di Veroli et al., 2016*). In total, we obtained 16,907 experiments for 583 combinations in 29 cell lines for further analysis. Drug targets of 38 monotherapy drugs were collected from the DrugBank database (*Wishart et al., 2018*) and the ChEMBL database (*Zdrazil et al., 2024*). The original names of all pathways mentioned in the manuscript are listed in *Supplementary file 1E*.

## Acknowledgements

This work was partially supported by funding from the Swedish Research Council (No. 2019-01857) and CancerFonden (22 2020 Pj), and the Swedish Foundation for Strategic Research (SSF). The computations were enabled by resources provided by the National Academic Infrastructure for Supercomputing in Sweden (NAISS), partially funded by the Swedish Research Council through grant agreement no. 2018-05973 and NAISS 2024/5-664. We acknowledge the investigators of the AZS DREAM Challenge for data access.

## Additional information

### Funding

| Funder | Grant reference number | Author |
| --- | --- | --- |
| Swedish Research Council | 2019–01857 | Trung Nghia Vu |
| Cancerfonden | 22 2020 Pj | Trung Nghia Vu |

The funders had no role in study design, data collection and interpretation, or the decision to submit the work for publication.

### Author contributions

Quang Thinh Trac, Data curation, Software, Formal analysis, Visualization, Writing – original draft, Writing – review and editing; Yue Huang, Data curation, Software, Formal analysis, Writing – original draft; Tom Erkers, Päivi Östling, Anna Bohlin, Albin Osterroos, Mattias Vesterlund, Rozbeh Jafari, Ioannis Siavelis, Helena Backvall, Santeri Kiviluoto, Lukas Orre, Janne Lehtiö, Soren Lehmann, Olli Kallioniemi, Writing – review and editing; Mattias Rantalainen, Methodology, Writing – review and editing; Yudi Pawitan, Conceptualization, Supervision, Methodology, Writing – review and editing; Trung Nghia Vu, Conceptualization, Resources, Data curation, Supervision, Investigation, Methodology, Writing – review and editing

### Author ORCIDs

Quang Thinh Trac ⓘ https://orcid.org/0000-0003-2429-0287
Mattias Vesterlund ⓘ https://orcid.org/0000-0001-9471-6592
Janne Lehtiö ⓘ https://orcid.org/0000-0002-8100-9562
Trung Nghia Vu ⓘ https://orcid.org/0000-0001-7945-5750

Reviewer #1 (Public review): https://doi.org/10.7554/eLife.100071.4.sa1
Reviewer #2 (Public review): https://doi.org/10.7554/eLife.100071.4.sa2
Reviewer #3 (Public review): https://doi.org/10.7554/eLife.100071.4.sa3
Author response https://doi.org/10.7554/eLife.100071.4.sa4

# Additional files

## Supplementary files
MDAR checklist

Supplementary file 1. Supplementary tables.

## Data availability
The implementation of DIPx, and related data are publicly available in https://www.github.com/trac-quangthinh/DIPx (copy archived at *Trac, 2025*). Drug synergy data are available from their original studies: Synapse database at https://doi.org/10.7303/syn4231880 for the AZS dataset, raw data from the supplementary data for the ONeil dataset at https://doi.org/10.1158/1535-7163.MCT-15-0843. The implementation of TAIJI-M as the molecular model is available at https://github.com/GuanLab/TAIJI/ (*Li, 2021*).

The following previously published dataset was used:

| Author(s) | Year | Dataset title | Dataset URL | Database and Identifier |
|---|---|---|---|---|
| Dry J, Guinney J, Saez-Rodriguez J, Norman T, Stolovitzky G, Friend S, Menden M, Wang W, Bare C, Tang EKY, Ghazoui Z, Barrett R, Edvardsson U, Vincent J, Garnett M, Bignell G, Forbes S, Esteller M | 2015 | AstraZeneca-Sanger Drug Combination Prediction DREAM Challenge | https://doi.org/10.7303/syn4231880 | Synapse, 10.7303/syn4231880 |

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
