## [Editor Report · eLife Assessment]

This **valuable** study presents a deep learning framework for predicting synergistic drug combinations for cancer treatment in the AstraZeneca-Sanger (AZS) DREAM Challenge dataset. The level of evidence seems **solid**, although performance on some datasets seems unconvincing and further validation would be required to demonstrate the generalizability of the model and, in turn, its clinical relevance. The reported tool, DIPx, could be of use for personalized drug synergy prediction and exploring the activated pathways related to the effects of drug combinations.

---

## [Referee Report · Reviewer #1 (Public review)]

The authors introduces DIPx, a deep learning framework for predicting synergistic drug combinations for cancer treatment using the AstraZeneca-Sanger (AZS) DREAM Challenge dataset. While the approach is innovative, I have following concerns and comments, and hopefully will improve the study's rigor and applicability, making it a more powerful tool in real clinical world.

(1) The model struggles with predicting synergies for drug combinations not included in its training data (showing only Spearman correlation 0.26 in Test Set 2). This limits its potential for discovering new therapeutic strategies. Utilizing techniques such as transfer learning or expanding the training dataset to encompass a wider range of drug pairs could help to address this issue.

(2) The use of pan-cancer datasets, while offering broad applicability, may not be optimal for specific cancer subtypes with distinct biological mechanisms. Developing subtype-specific models or adjusting the current model to account for these differences could improve prediction accuracy for individual cancer types.

(3) Line 127, "Since DIPx uses only molecular data, to make a fair comparison, we trained TAJI using only molecular features and referred to it as TAJI-M.". TAJI was designed to use both monotherapy drug-response and molecular data, and likely won't be able to reach maximum potential if removing monotherapy drug-response from the training model. It would be critical to use the same training datasets and then compare the performances. From Figure 6 of TAJI's paper (Li et al., 2018, PMID: 30054332) , i.e., the mean Pearson correlation for breast cancer and lung cancer are around 0.5 - 0.6.

The following 2 concerns have been included in the Discussion section which are great:

(1) Training and validating the model using cell lines may not fully capture the heterogeneity and complexity of in vivo tumors. To increase clinical relevance, it would be beneficial to validate the model using primary tumor samples or patient-derived xenografts.

(2) The Pathway Activation Score (PAS) is derived exclusively from primary target genes, potentially overlooking critical interactions involving non-primary targets. Including these secondary effects could enhance the model's predictive accuracy and comprehensiveness.

---

## [Referee Report · Reviewer #2 (Public review)]

Trac, Huang, et al used the AZ Drug Combination Prediction DREAM challenge data to make a new random forest-based model for drug synergy. They make comparisons to the winning method and also show that their model has some predictive capacity for a completely different dataset. They highlight the ability of the model to be interpretable in terms of pathway and target interactions for synergistic effects.

In their revised manuscript and response, the authors have tried to address all points. I do not fully agree with them about the definition of overfitting still. If the objective it to identify synergies given any 2 drugs, not just those in a dataset at different doses, then the results certainly appear overfit to the training set given the performance degradation. However, at this time, I cannot add any useful suggestions to improve performance.

---

## [Referee Report · Reviewer #3 (Public review)]

Summary:

Predicting how two different drugs act together by looking at their specific gene targets and pathways is crucial for understanding the biological significance of drug combinations. This study incorporates drug-specific pathway activation scores (PASs) to estimate synergy scores as one of the key advancements for synergy prediction. The new algorithm, Drug synergy Interaction Prediction (DIPx), developed in this study, uses gene expression, mutation profiles, and drug synergy data to train the model and predict synergy between two drugs. Comprehensive comparisons with another best-performing algorithm, TAIJI-M, highlight the potential of its capabilities.

Strengths:

DIPx uses target and driver genes to elucidate pathway activation scores (PASs) to predict drug synergy. Its performance was tested using the AstraZeneca-Sanger (AZS) DREAM Challenge dataset, especially in Test Set 1, where the Spearman correlation coefficient between predicted and observed drug synergy was 0.50 (95% CI: 0.47-0.53). DIPx's ability to handle novel combinations, as evidenced by its performance in test set 2, indicates the potential for predicting new and untested drug combinations, even though it's lower than that of the test set 1.

Weaknesses:

While the DIPx algorithm shows promise in predicting drug synergy based on pathway activation scores, it's essential to consider its limitations. One limitation is that the availability of training data for specific drug combinations may influence its predictive capability. Further testing and experimental validation of the predictions in future studies would be necessary to assess the algorithm's generalizability and robustness.

---

## [Author Response]

The following is the authors’ response to the previous reviews

We would like to respond to just one remaining concern from Reviewer 1 and Reviewer 2 regarding a potential overfitting in Test Set 1, which involves combinations already present in the training set. DIPx’s (and TAIJI’s) performance in Test Set 1 is better than in Test Set 2, which involves combinations not present in the training set. Let’s consider two general points to highlight why the improved performance is not the result of overfitting.

(1) Suppose we are testing the e ect of one drug D; the training may involve, for example, selecting an optimal dose. A validated e ect of D in an independent test set is not an overfit, even though we are using the same drug in the training and the test set. Testing one drug is an extreme case, but the same idea holds for any number of drugs. What matters is the independence of the test set.

(2) A prediction model P1 will legitimately perform better than model P2, if P1 uses better or more informative features than P2. The features could be those used directly in the model, but they could also be other observable characteristics not directly used in the model, such as optimal subregions of the feature space. DPIx or TAIJI results indicate that the identity of previously trained combinations is one such informative feature. The set of previously trained combinations corresponds to a subregion of the feature space. DIPx’s prediction performance for known combinations would be expected to follow the results from Test Set 1; we cannot expect that if there is an overfitting issue. Finally, we note that Test Set 1 was established and used in the AstraZeneca Dream Challenge for rigorously testing the prediction of known combinations.